

# Reconstructing Central African hydro-climate over the past century using freshwater bivalve shell geochemistry

Zita Kelemen[1], David P. Gillikin[2], Steven Bouillon[1]

[1] Department of Earth & Environmental Sciences, KU Leuven, Celestijnenlaan 200 E, B-3001 Leuven, Belgium
5   [2] Department of Geosciences, Union College, 807 Union St., Schenectady, NY 12308, USA

*Correspondence to*: Zita Kelemen (zitakelemen2@gmail.com), Steven Bouillon (steven.bouillon@kuleuven.be)

**Abstract.** Centennial records of climatic and hydrological data are scarce on the African continent. Freshwater bivalve shells have the potential to record climate-related proxies, from which reconstructions of e.g., river discharge or water isotope variations over long time periods would be possible. The Oubangui River, the largest right-bank tributary of the Congo River, 10   is one of the few African rivers for which long-term discharge records are available. This, together with the availability of museum-archived shells, makes it an ideal location to study changes in hydroclimate in central Africa over the past century and to validate the potential of proxies recorded in freshwater shells. We analysed the carbon and oxygen isotope composition ($\delta^{13}C$, $\delta^{18}O$) across the growth axis of museum-archived (collected between 1891 and ~1952) and contemporary (live collected in 2011 and 2013) *Chambardia wissmanni* shells from the Oubangui River, covering sections of the past ~120 years. Shell 15   isotopes show a clear baseline shift. Both $\delta^{13}C$ and $\delta^{18}O$ exhibit pronounced cyclicity in recent and historical shells, but recent shells showed a much wider range in $\delta^{18}O$ values and a narrower range in $\delta^{13}C$ values compared to historical specimens. The historical $\delta^{18}O_{shell}$ values covered only the lower part of the range measured in recent shells, which suggests a major change in the low flow conditions of the Oubangui River between the 1950s and 2010s. Reconstructed discharge values, based on the logarithmic relationship established between recent water $\delta^{18}O$ values and measured discharge, tended to overestimate the low 20   flow values in the past, suggesting a different $\delta^{18}O_w$ - discharge relationship in the earlier part of the 20th century. Thus, while the freshwater shell $\delta^{18}O$ patterns are consistent with the documented long-term discharge changes in the Oubangui, the shells show that the most pronounced changes in hydroclimate over the past century are expressed in low-flow sections of the hydrograph, and likely result from a combination of changes in the rainfall regime, aquifer recharge, and/or vegetation cover in the upper catchment. These results are consistent with the suggestion that dry periods in the upper Congo basin are becoming 25   more extreme in recent times, and highlight that freshwater shells could offer a valuable archive to study recent changes in catchments where no long-term empirical hydrological or climatological data are available.

## 1 Introduction

The Oubangui River is the largest right-bank tributary of the Congo River and represents one of the few sites within the Congo Basin where discharge has been monitored since the early 20th century. Several studies have demonstrated that over the past



century, the annual average discharge at Bangui has decreased by 40%, while the dry season flow has decreased more

drastically, by more than 60% (Wesselink and Orange, 1996; Orange et al., 1997; Laraque et al., 2001; Runge and Nguimalet,

2005). Given the importance of river navigability for a landlocked country such as the Central African Republic, this decrease

in discharge has important ramifications for transport: while the river was navigable for larger boats for 8 months of the year

before the 1970's, this is currently limited to 4-5 months (Van Pul, 2023). Despite these substantial and well-documented

changes, rainfall in the watershed of the Oubangui River has decreased by only 5% since 1970 (based on 11 stations in the

Central Oubangui catchment; Nguimalet and Orange, 2015; Laraque et al., 2020).

Anthropogenic changes within a catchment, such as forest clearing and land-use change, water abstraction for irrigation have

often been suggested as driving forces of hydrological changes (Descroix et al., 2012; Recha et al., 2012; Tshimanga and

Hughes, 2012; Amogu et al., 2014; Essaid and Caldwell, 2017). There are signs that the entire Congo River basin is

experiencing a drying trend over the past decades, especially the northern part of the forest (Asefi-Najafabadi and Saatchi,

2013; Zhou et al., 2014; Malhi and Wright, 2004). However, land use in the Oubangui watershed upstream of Bangui has

changed relatively little over the past century according to Nguimalet and Orange (2013), and it is therefore currently not

considered to play a major role in explaining the decreased discharge.

Longer-term records of climatic and hydrological data are relatively scarce for the African continent: the data collection

network in the Congo basin is very sparse and few hydrological stations are still operational, with only a handful of sites for

which long-term data (i.e. spanning the past century) are available (Laraque et al., 2001; Alsdorf et al., 2016). The application

of proxies in biological archives therefore has a high potential to reconstruct environmental records over the last

decades/centuries. There is an increasing set of studies from various lacustrine environments (Barker et al., 2011; Berke et al.,

2012; Mologni et al., 2024)  and tree cores (Battipaglia et al., 2015; Van Der Sleen et al., 2015), but freshwater bivalve shells

have so far been rarely used on this continent (Abell et al., 1996; Kelemen et al., 2017, 2021). However, a number of studies

in temperate regions have successfully used freshwater bivalve shells to shed light on past changes in freshwater ecosystems

(Dettman and Lohman, 2000; Dettman et al., 1999, 2001, 2004; Pfister at al., 2018, 2019; Schöne et al., 2020; Versteegh et

al., 2010, 2011).

Bivalve shells are considered to be useful biological archives to study sub-seasonal changes in aquatic systems (reviewed in

Stringer and Prendergast, 2023). Their abundance, filter-feeding lifestyle and sequentially secreted carbonate shell makes them

ideal archives for reconstructing water isotope ratios, as confirmed by Pfister et al. (2019) although the latter review focused

on average values and did not focus on higher-resolution (temporal) data. However, over different time scales, freshwater

bivalves have been shown to record changes in discharge and precipitation (Ricken et al., 2003; Dettman et al., 2004; Kaandorp

et al., 2005; Versteegh et al., 2011; Kelemen et al., 2017, 2021), as they precipitate their shell carbonate in oxygen isotopic

equilibrium with the host water (Grossman and Ku, 1986; Dettman et al., 1999; Goodwin et al., 2019).

With the exception of *Margaritifera* species with a reported maximum lifespan of 200 years or more (see Stringer and

Prendergast, 2023), most freshwater bivalve species have a relatively short lifetime (~1 decade), which limits the period of



environmental data recorded by individual specimens. Sampling the sequentially secreted layers at high spatial resolution can reveal detailed information across several hydrological cycles. We have demonstrated earlier that the bivalve *Chambardia wissmanni* precipitates its aragonitic shell in oxygen isotope equilibrium with the host water in the Oubangui and Niger River (Kelemen et al., 2017). Seasonal temperature variations in such tropical rivers are relatively limited and thus, oxygen isotope ratios in bivalve shells ($\delta^{18}O_{shell}$) are predominantly influenced by the oxygen isotope ratios of water ($\delta^{18}O_w$) (Kelemen et al., 2017), which in turn are predominantly controlled by rainfall patterns and evaporation (Gat, 1996).

In this study, we aimed at reconstructing aspects of the hydroclimate of Central Africa using high-resolution stable isotope analyses ($\delta^{13}C$ and $\delta^{18}O$) on both recent and archived freshwater shells from the Oubangui River. The choice for the Oubangui River stems from the fact that (i) we have previously calibrated isotope proxies in bivalve shells from this river by comparing shell data with in situ water data collected during the shell growth period (Kelemen et al., 2017), (ii) shell specimens were available from museum collections that spanned a wide period from late 19th century to mid-20th century, and (iii) clear changes in the discharge of the Oubangui have been documented over part of the period covered by our shell collection, providing an independent verification of the applicability of our approach.

Moreover, regular sampling of river water over several years has shown that a clear negative relationship exists between discharge and $\delta^{18}O_w$ values (Kelemen et al., 2017). Thus, we hypothesized that historical specimens (pre-1960) would (i) record the documented long-term decrease in dry season and annual discharge, (ii) show a less pronounced amplitude in $\delta^{18}O_{shell}$ values, and more specifically lack the high $\delta^{18}O_{shell}$ values associated with low discharge conditions characteristic of the most recent decades (as shown by Kelemen et al., 2017).

## 2. Materials and Methods

### 2.1 Oubangui basin

The Oubangui River (Fig. 1) is the largest right-bank tributary of the Congo River, with an average annual discharge measured at Bangui (Central African Republic) of 3700 m³s-1 (for the period of 1935-2015; Nguimalet and Orange, 2015) and a length of 2400 km including its upper reaches, the Uele River (Boulvert, 1987). Along its path to the Congo River, the Oubangui mainly passes through peneplain landscapes, relatively flat open landscapes in the North, mosaics of forest – savannah in its central basin, and transforms into dense humid rainforest along its southern reaches (Mayaux et al., 1999). Influenced by the seasonal migration of the equatorial (tropical) rain belt, the highest rainfall is during the July – October period in the central part of the basin, while closer to the equator, the rainfall peak is bimodal, with the highest peak during September to December and March to May (Mahé, 1993). The central Oubangui River watershed is entirely within the Central African Republic (CAR), as in this part of the basin tributaries are joining exclusively from the right bank. The general water recharge in the central Oubangui is supplied by abundant rainfall north of the main stem during the wet season. In the CAR, numerous rain-gauging stations have been established in the past century, however few stations have continuous (multi decadal) measurements



available, and even less with data after the 1980s (data can be found in the SIEREM database, provided by the HydroSciences
Montpellier    Laboratory,    at    http://www.hydrosciences.fr/sierem/index_en.htm).    Continuous    precipitation    isotope
measurements are only available for two stations at Bangui, for the period 2009 – 2021, provided by IAEA (Global Network
of Isotopes in Precipitation (GNIP; available via https://nucleus.iaea.org/sites/ihn/Pages/GNIP.aspx). The single peak
discharge period at Bangui results from a transitional tropical regime of the basin, with one wet and one dry season, and annual
precipitation of 1475 mm (average over the period between 1935 and 2015 at Bangui by Nguimalet and Orange, 2019). The
highest discharge is usually measured between September and November, when the rainy season peak arrives from the upper
catchment, and water levels are the lowest after the dry season between February and April. The gauging station at Bangui is
one of the few sites within the Congo Basin for which a long−term discharge record is available (Orange et al., 1997; Wesselink
and Orange 1996; Laraque et al., 2001; Runge and Nguimalet, 2005), and this station provides discharge data from a catchment
area of about 490,000 km$^2$ upstream of Bangui. The discharge monitoring station was established in 1911 at Bangui, but a few
years of intermittent records follow, leaving a relatively long data gap between 1920 and 1935, after which continuous daily
measurements are available. Discharge data were obtained from the Direction de la Météorologie Nationale (CAR).

## 2.2 Shell collection and analyses

Recent specimens of *C. wissmanni* were collected in the Oubangui River in Bangui between 2011 and 2014 (see Kelemen et
al., 2017). We searched museum collections (initially from the MusselP database; http://mussel-project.uwsp.edu) for
specimens of *C. wissmanni* that could unambiguously be assigned as originating from the Oubangui River, and with sufficient
information on the collection date or period. A total of eight shells (see Table 1 for details) could be sourced from various
museums (MRAC-Royal Museum for Central Africa, Tervuren, Belgium; IRSNB-Royal Belgian Institute of Natural Sciences,
Brussels, Belgium, and MNHN-Muséum National d'Histoire Naturelle, Paris, France). Two *C. wissmanni* specimens (labeled
as *Spatha rubens*) were collected in 1891 by Jean Dybowski, based on the mission reports one of these shells (1891A – August
1891) was collected a few km above the confluence of Oubangui and Congo Rivers, while the specimen from December 1891
(1891D) was collected at Bangui. One *C. wissmanni* shell from 1904 mentions that it was collected between 100 and 250 km
downstream from Bangui, somewhere between Libenge and Dongo (Fig. 1). The 1908 specimen (labeled as *Spathopsis
wissmanni*) was collected ~500 km downstream of Bangui at Longo during the Robert Hottot mission. Two shells sampled in
1914 (labeled as *Spatha oubanghi*) are from Bangui (1914A, 1914B). Two bivalves were collected downstream of Bangui,
120   near Mawuya (*Aspatharia*) and Dongo (*C. wissmanni*). The specimen from Mawuya is part of a larger, carefully preserved
collection (different species, all valves paired and in excellent condition, each pair bound together with string), thus even
though we could not find the exact date of sampling we strongly expect late 1947 or early 1948 to be the collection period, as
two large fauna collection campaigns were held in Mawuya by the Royal Belgian Institute for Natural History - in October
1947 and in January-February 1948. *C. wissmanni* from Dongo was a part of a collection of a known collector, R.H.
125   Raemackers; this specimen entered the collection on November 9th, 1957, yet the exact date of collection is not provided – we



therefore considered the shell likely to have been from the 1950s. The shell collection was reviewed by Graf and Cummings (2011) and all specimens we used were confirmed to be *C. wissmanni*.

Shells were sectioned along the maximal growth axis and sections of few mm thickness were mounted on glass slides. The prismatic layer was sampled from the tip towards the umbo using a New Wave Micromill equipped with 300 μm diameter drill bit. The sampling resolution was between 350-750 μm and 300-450 μm depth, resulting in 50 to 80 μg powder for analysis. Samples were collected in 12 mL Labco Exetainers, flushed with Helium and reacted with >100% phosphoric acid. The evolved carbon dioxide was then analyzed on a Thermo Delta V Advantage isotope ratio mass spectrometer coupled to a GasBench II, either at Union College (NY, USA) or KU Leuven (Belgium). Along with the samples, certified reference materials (LSVEC, NBS−18, NBS−19) and in-house $CaCO_3$ standards were analyzed and the long-term standard deviations were better than 0.1‰ for both $\delta^{13}C$ and $\delta^{18}O$.

To calculate reconstructed $\delta^{18}O_w$ values, the 1000Ln(α) equation of Dettman et al. (1999) (based on data from Grossman and Ku, 1986) was used (Eq. 1, 2), including the conversion factor from the VSMOW to the VPDB scale of 1.0309 (Gonfiantini et al., 1995).

$$1000\ln(\alpha) = 2.559 \, (106 \, T^{-2}) + 0.715 \qquad \text{[equation 1]}$$

where T is the water temperature in Kelvin and α is the fractionation between water and aragonite described by the following equation:

$$\alpha(\text{aragonite-water}) = (1000 + \delta^{18}O_{ar} \, (\text{VSMOW})) \, / \, (1000 + \delta^{18}O_w \, (\text{VSMOW})) \qquad \text{[equation 2]}$$

For further calculations an average temperature of 28.6 ºC (derived from in situ measurements over three years of monitoring; see Kelemen et al., 2017) was used, assuming only a minimal water temperature change over the past century, based on the measured centennial air temperature increase in the 20th century of about 0.5 ºC on the African continent (Hulme et al., 2001). *Chambardia wissmanni* is known to precipitate aragonite shells (all Unionids make aragonite shells; Strayer and Malcom 2007). Experimental studies (e.g., Casella et al., 2017) have shown that metastable aragonite does not transform to more stable structure of calcite below 175 ºC, especially over the time frames considered here. Fossil freshwater bivalves (*Diplodon longulus*) collected from the Miocene Pebas Formation in Amazonia showed no evidence of diagenetic alteration (Kaandorp 2005; Vonhof et al., 1998, 2003), nor did 66 Ma (Cretaceous) freshwater shells from the Hell Creek formation (Dettman and Lohmann 2000, Tobin et al., 2014), thus we can be confident that the aragonite structure of our collection stored for ~100 years at ambient conditions is unaltered.

The full set of $\delta^{13}C$ and $\delta^{18}O$ data on historical shells is provided as an electronic supplement. Data on recent shells (Kelemen et al., 2017) are publicly accessible via ResearchGate (https://www.researchgate.net/publication/331716269).



## 3. Results

The $\delta^{18}O_{shell}$ cyclicity in the historical specimens was well defined, with values ranging between -5.0 and -1.4 ‰ and individual shell amplitudes (difference between maximum and minimum values) ranging between 1.6 and 2.7‰ (Fig. 2 and 3). In recent

shells, $\delta^{18}O_{shell}$ values ranged between -5.1 and +0.2 ‰, with amplitudes up to 5.0 ‰, considerably higher than the historical specimens. With only a few exceptions, the $\delta^{18}O_{shell}$ data in historical specimens remained below -2 ‰, whereas the modern shells were often above -1 ‰ (Fig. 3). In the collection of historical specimens there was no considerable difference in $\delta^{18}O_{shell}$ record between the bivalves collected at Bangui and those collected somewhat more downstream (Fig. 2).

$\delta^{13}C_{shell}$ values in historical specimens ranged between -15.7 and -7.0‰ and show clear cyclicity (Fig. 4). They were non-

synchronous with $\delta^{18}O_{shell}$ cycles, but the patterns observed in the older shells were analogous with the recently collected specimens (Kelemen et al., 2017). The $\delta^{13}C_{shell}$ range was slightly wider than that in modern specimens (from -14.6 to -8.2‰; Kelemen et al., 2017). With one exception (a young specimen from 1904), all shells expressed an ontogenetic decrease of $\delta^{13}C_{shell}$ values (Fig. 4; but note that the modern shells are a compilation of many shells while the older shells represent individuals).

## 4. Discussion


Freshwater bivalve shells spanning the past ~120 years show a major hydrological shift in the northern Congo basin over this time frame. Unionid bivalves precipitate shell material in oxygen isotope equilibrium with the surrounding water, thus reflecting $\delta^{18}O$ values of the water and temperature (Dettman et al., 1999; Versteegh et al., 2011; Kelemen et al., 2017; Goodwin et al., 2019), but in these tropical specimens, temperature plays a minor role, as the temperature variations throughout

the year are relatively low (28.6 ± 1.2 °C for continuous measurements between 2010-2013; discussed in Kelemen et al., 2017). Thus, the difference between the historical and recent bivalves clearly reflects a difference in $\delta^{18}O_w$ between the early-mid 20th century and those of the past decade (i.e. 2005-2013). This matches with the expectation that shells recorded the different discharge conditions from the early 20th century, but a robust interpretation requires a more detailed look at the data and assumptions. Therefore, in the following discussion we will first briefly describe the most important characteristics of the

Oubangui River, then subsequently attempt to reconstruct discharge by using the shell $\delta^{18}O$ record, and discuss whether a change in the $\delta^{18}O$-discharge relationship might have occurred over time. We then explore the possible cause(s) of the lower $\delta^{18}O$ values of historical shells in comparison with the recently collected specimens in the context of shell collection sites, rainfall anomalies, and land-use/vegetation changes.



## 4.1 The Oubangui River discharge (Q)

The Oubangui River has a single peak discharge regime, and the century-long discharge record showed a stable phase since the beginning of the record until the early 1960's, when a decade of wetter conditions was experienced in Central and West Africa, which was followed by a hydrological deficit over the entire basin (Orange et al., 1997, Laraque et al., 2001, 2013, 2020). The 100-year flood was measured in October 1916, when a discharge of 14,000 $m^3 s^{-1}$ was recorded, and the lowest discharge was measured in April 1990 (266 $m^3 s^{-1}$) (Fig. 5). In the centennial Q record, a difference can be seen between the flow in the early-mid 20th century, when the lowest Q was generally not below 700 $m^3 s^{-1}$ (with three exceptional years: 1914, 1945 and 1946), while after the 1970s the low discharge gradually decreased, regularly reaching values between 250-500 $m^3 s^{-1}$ during the dry season. However, the decrease in peak discharge is also substantial over this time period (pre1970 ≈ 10-14,000 $m^3 s^{-1}$, recent ≈ 7-9,000 $m^3 s^{-1}$) (Fig. 5).

## 4.2 Discharge reconstruction based on shell δ18O

As determined by Kelemen et al. (2017), the Oubangui River at Bangui exhibits a logarithmic relationship between $\delta^{18}O_w$ and Q ($\delta^{18}O_w$=-1.222ln(Q)+8.741; $R^2$=0.75). Due to the logarithmic nature of this relationship, the most robust discharge reconstructions based on $\delta^{18}O_w$ are obtained for low discharge values (when the correlation is nearly linear), while for higher discharge the data are more scattered, making the reconstruction less accurate. Using $\delta^{18}O_{shell}$ values from archived bivalves and measured river discharge, we investigated if the $\delta^{18}O_w$ - Q relationship changed over time, thereby indicating changes in hydroclimate. The availability of bivalve shells collected at Bangui that cover suitable time frames was limited; only the very beginning of the continuous discharge data series is covered by shells collected at Bangui, as bivalves from Bangui were mainly collected before 1911. The other shells covering most of the record were collected downstream from Bangui. Although not all shells were collected at Bangui, they show similar $\delta^{18}O_{shell}$ profiles suggesting limited influence of downstream inputs on river water isotope values (Fig. 2).

The shells record a large shift in dry season river water $\delta^{18}O$ values and thereby suggest a major change in watershed hydrology and/or hydroclimate. Dry conditions and low flow are reflected in shells as higher $\delta^{18}O$ values. Thus, if we assume as a first approach that the $\delta^{18}O_w$-Q relationship measured between 2009-2013 (Kelemen et al., 2017) can be considered as representative for the full timeframe of our shell collection, substantially lower peak $\delta^{18}O_{shell}$ values in historical specimens suggest that discharge was substantially higher during low flow conditions at Bangui in the past. The $\delta^{18}O_{shell}$ record also suggests that the $\delta^{18}O_w$ values characterizing the high discharge remained similar over the past 100 years (based on $\delta^{18}O_{shell}$ minima being similar through time (Fig. 3)). Constant $\delta^{18}O_w$ values over the past century would suggest that the hydroclimatology of the high flow period has not changed considerably.

A large decrease has been observed in the discharge record during the 1970s (Fig. 5), however, high and low flows were not evenly affected: peak flow decreased 35%, while the flow minima decreased 60% (Orange et al., 1997). Flow minima are indeed thought to be more sensitive to hydroclimatological changes than high (or annual) flows (Wesselink and Orange 1996;



Bricquet et al., 1997; Orange et al., 1997). This might be explained by the different origin of water during high and low flow periods: while peak flow is fed by surface runoff directly provided by precipitation, baseflow (low flow during dry season) is maintained by riparian aquifers and water stored in upstream channels (Delleur, 1999; Brutsaert, 2005). Therefore, we
hypothesize this different response of discharge phases during the 1970s, when changes in the sources of water feeding the river (precipitation, groundwater), likely also occurred.

The lower amplitude and generally lower $\delta^{18}O_{shell}$ values in historical shell specimens are qualitatively consistent with the trends in long-term discharge data, and extend this trend into the late 19th century for which discharge data are not available. However, it must be kept in mind that the Q-$\delta^{18}O_w$ relationship might vary over time, and that the one established based on
recent data might not be applicable to the historical data. To explore this possibility, we compared the reconstructed discharge data (using the recent Q- $\delta^{18}O_w$ relationship) with the empirical discharge record, which indeed indicates that low discharge is consistently overestimated in the older shells (Fig. 6). In addition, the reconstructed high discharge record exceeds the discharge above which *C. wissmanni* seems to temporarily stop precipitating carbonate (Kelemen et al., 2017). It should be kept in mind however, that the shell from 1948 was collected downstream from Bangui, and this may affect the Q-$\delta^{18}O_w$
relationship locally (as discussed further on). Nevertheless, the 1914 shell, collected at Bangui, appears to overestimate dry season discharge, and thus our data indicate that the Q- $\delta^{18}O_w$ relationship might indeed have changed through time.

**4.3 Influence of vegetation cover, precipitation and groundwater on $\delta^{18}O_w$ values**

In order to explore possible mechanisms leading to a proposed difference in the $\delta^{18}O_w$ -Q relationship between recent direct observations and that based on pre-1960 bivalve shell data, several factors should be considered. We first examine whether a
change in precipitation could influence the relationship, and whether a shift in dry season $\delta^{18}O_w$ values is more likely to be caused by large scale or by more regional-scale changes in precipitation and evapotranspiration. Other possibilities are also explored, such as the possible influence of the shell collection site (distance from Bangui along the Oubangui River), possible impacts of changing vegetation cover and of groundwater fluctuations.

**Influence of precipitation**

Specimens collected before 1960 showed a generally smaller $\delta^{18}O_{shell}$ range and recorded only the lower $\delta^{18}O$ values in comparison with the shells collected after 2010 (Fig. 3), which suggests a substantially higher dry season river flow (with correspondingly lower $\delta^{18}O_w$ values) during the first part of the 20th century. However, it is also possible that substantial changes in the precipitation regime, and thus $\delta^{18}O$ values, have occurred in the region during this period.

In the centennial rainfall record from Bangui, a large decrease in yearly rainfall amount was observed following the wet decade
in the 1960s (Orange et al., 1997; Nguimalet and Orange, 2013, 2019). This precipitation regime shift was noticed throughout central Africa and over a major part of West Africa as well. The year 1970 is most commonly accepted as being the beginning of a major drought in Sub-Saharan Africa, with a precipitation decrease up to 40% (Mahé and Olivry 1999; Mahé et al., 2001;



Hulme et al., 2001; Dai et al., 2004). According to Nguimalet and Orange (2019), statistical calculations indicate a decrease in rainfall in the Oubangui basin for the period 1935-2015 only in 1970. The measured rainfall decrease was about 5%, which
cannot directly explain the annual average discharge decrease of 22% for the 1982-2013 period at Bangui (Nguimalet and Orange, 2019). However, these studies focused on yearly averaged rainfall data - we refined this analysis for two stations with a continuous multi-decadal monthly rainfall record in the Oubangui basin upstream of Bangui: Bangui (1907-2007) and Bambari (1931-2007). When monthly data were separated into dry season (December, January, February) and wet season (remaining months), it becomes apparent that the amount of precipitation during wet months did not change notably at either
of the stations (Fig. 7). Conversely, precipitation during dry months (in particular December and January) gradually decreased at Bambari starting after 1970. Moreover, rainfall in March at Bambari shows a decreasing trend since 1982, which suggests that the dry season may have increased in length in this area.

Changes in precipitation amount, air mass trajectory, and temperature lead to changes in $\delta^{18}O_{precip}$ values (Gat, 1996). If during the dry season less rainfall occurs, the precipitation-evaporation balance shifts towards evaporation and river water $\delta18O$
values are expected to increase. Indeed, rainfall is usually more depleted in $^{18}O$ in comparison with the water source, because the water molecules with lighter isotopes evaporate faster, thus the residual evaporating source becomes more enriched. At times of heavy rainfall, rain is usually very depleted in $^{18}O$ as a result of more complete rainout of water vapor, a process referred to as the "amount effect" (Dansgaard 1964). In contrast, dry season precipitation is typically $^{18}O$-enriched in comparison to that during the wet season (Dansgaard 1964; Gat 1996; Balagizi et al., 2019). The precipitation $\delta^{18}O$ record at
Bangui indeed shows the "amount effect", during abundant rainfall the $\delta^{18}O$ values are lower than $\delta^{18}O$ values of low rain amounts (Figure 8). However, wet and dry season precipitation data are not perfectly separated in terms of isotope composition. All these factors lead to an increase of $\delta^{18}O_w$ values during the dry season, which was clearly recorded in the recently collected shells.

**Influence of the collection site and vegetation cover**

The Oubangui Basin has a diverse landscape, the central region where the river receives only right bank tributaries is covered by tree savannahs, while the lower section (downstream from Bangui) flows through rainforests. Not all museum archived shells were collected near Bangui, and river location is important to consider in interpreting environmental signals, as it is possible that the specimens collected downstream of the confluence with the Lobaye (shells 1904, 1948, 1950s, 1908, and 1891A, see Fig. 1, 2) may be influenced by inputs from tributaries that drain rainforests. The Lobaye River represents the
boundary between savannah and rainforest in the basin (see Bouillon et al., 2014; Kelemen, 2019). The Lobaye and its right bank tributaries flow through the humid Ngotto rainforest, which is considerably different from most of the upper Oubangui watershed. The Ngotto rainforest likely has dry season precipitation with lower $\delta^{18}O_w$ values compared to the Oubangui watershed. The Lobaye River has a relatively stable discharge throughout the year, with a range on average between 240 and 500 m³s-1, and a Qmax/Qmin about 2 for the available record (1950-1987) at the Mbata gauge station. Given the low variation



in discharge, δ18Ow values are expected to be similarly uniform, thus the very few available measured river $\delta^{18}O_w$ values ranged between -3.0 and -3.3‰ (n=13, November 2012; Kelemen 2019) are most likely representative (similar to $\delta^{18}O$ measured in a single rain event on 20 Nov. 2012: -1.9‰). During the low discharge period, the Lobaye River provides nearly 50% of the flow of the lower Oubangui River, which isotopically should reflect Ngotto rainforest precipitation. While in the two bivalves collected just below the confluence with Lobaye (shell 1904 and 1948), $\delta^{18}O$ values occasionally reached values

somewhat higher than in historical bivalves collected more upstream, most of their $\delta^{18}O$ record fell in the same range.

The two shells collected near the confluence with the Congo River (shell 1908 and 1891A) showed a very narrow range of $\delta^{18}O_{shell}$ values, covering only the lowest $\delta^{18}O_{shell}$ values recorded in the other shells. This corroborates the hypothesis that the three rainforest rivers (Ibenga, Motaba and Ngiri) contribute to more steady and lower $\delta^{18}O_w$ values in the lower Oubangui (Fig. 2). Nevertheless, the older shells collected at, or upstream of Bangui (shells 1914A, 1914B, and 1891D) all have similar

isotope profiles to shells collected downstream of the Lobaye. This is particularly evident by direct comparison of shells 1891D and 1891A, which were both collected in the same year, but grew in different sections of the river: 1891D close to Bangui and 1891A ~500 km downstream (Fig. 2). This suggests that the archived shells do indeed reflect dry-season precipitation with lower $\delta^{18}O$ values in the Oubangui and that the long-term trend in $\delta^{18}O_{shell}$ record is not skewed due to some shells having been collected downstream of rainforest watersheds.

**Influence of groundwater**

Orange et al. (1997) noted the sponge-like functioning of the majority of the Oubangui Basin, based on the three year time lag between the decrease in rainfall and the change in the discharge record. This behaviour of the basin might be the main reason why the discharge decline at Bangui was so intense and rapid after 1970, while the rainfall record showed only minor changes with a sudden reduction of precipitation in 1968 (Nguimalet and Orange, 2013, 2019). Moreover, a study of the Oubangui at

Mobaye (upstream of Bangui) showed that at the northern tributary (Kotto), which exclusively drains savannah, had its infiltration capacity to recharge the aquifer altered by the long drought (Nguimalet et al., 2022). This would result in the abundant precipitation being diverted from recharge to runoff, especially after the drought of the 1970s (Wesselink & Orange 1996; Bricquet et al., 1997). The drier period starting around 1970 would thus have led to a decrease in groundwater storage, which is currently not high enough to sustain the hydrological regime as before during the dry season (Orange et al., 1997). In

contrast, the southwest of the basin (downstream from Bangui) lies above a semi-continuous aquifer which contributes to river discharge during low flow (Laraque et al., 2001), and might explain why the Oubangui at Bangui remained in drought phase while the Congo River at Brazzaville returned into its normal flow conditions after the 1990s (Laraque et al., 2013).

Jasechko & Taylor (2015) found that groundwater stable isotope values across different tropical catchments typically followed that of annual precipitation, weighted toward more rainfall-intense months. As the aquifer is recharged by rainwater (in

particular during wet periods), it preserves the $\delta^{18}O$ values of the precipitation. Thus during low flow, when the groundwater is the main supply of the river, $\delta^{18}O_w$ values are expected to reflect the $\delta^{18}O$ values of the annual precipitation. If that principle



is applicable to our study basin, this would suggest that groundwater $\delta^{18}O$ values would be lower than -2.3 ‰ (i.e. the amount-weighted average precipitation value at Bangui, see Fig. 8). Indeed, groundwater $\delta^{18}O$ values in shallow (3-20m deep) wells reported for Bangui range between -3.3‰ and -2‰ (Djebebe-Ndjiguim et al., 2019), while in the region of Mambéré-Kadéi

(region drained by the Lobaye River) west of Bangui, measured borehole water $\delta^{18}O$ values range between -3.5 and -2.4 ‰ (Foto et al., 2019).

These groundwater $\delta^{18}O_w$ values are in good agreement with the highest $\delta^{18}O_{shell}$ values of the historical specimens, as all showed low peak $\delta^{18}O$ values during low discharge in comparison with the $\delta^{18}O_{shell}$ values in historical specimens. This favours the hypothesis that a (partial) change in source and composition of the low flow water occurred. Under this hypothesis, the

Oubangi receives a decreased groundwater contribution (low $\delta^{18}O$ values) during low discharge since the drought in the 1970s, with the majority of the water sourced from the upper reaches of the river, where the water travels hundreds of kilometres through savannah, and from [18]O enriched precipitation. According to Nguimalet and Orange (2019), the volume mobilized by the Oubangui aquifer was gradually decreasing since 1970, in some years the water provided by the aquifer was less than 50% of the amount before the drought. Dry season precipitation with higher $\delta^{18}O$ values and without the dilution of groundwater,

would then be consistent with shells recording increased $\delta^{18}O$ values.

## 4.4 Shell $\delta^{13}C$ values

Some reports postulate that shell $\delta^{13}C$ values track $\delta^{13}C$ of the dissolved inorganic carbon (DIC) in the host water (Fritz & Poplawski, 1974; Poulain et al., 2010; Graniero et al., 2017). However, several authors found disequilibrium between DIC and freshwater shell $\delta^{13}C$ values (Dettman et al., 1999; Gillikin et al., 2009; Kelemen et al., 2017, Graniero et al., 2021), while

some studies reported constant fractionation (Abell and Hoelzmann, 2000; Kaandorp et al., 2003). There is a strong inverse relationship between discharge and $\delta^{13}C_{DIC}$ values in the Oubangui today (Bouillon et al., 2012, 2014). Bivalves collected in the 1950s and earlier exhibited pronounced cyclicity, with an amplitude between 5 and 7.9 ‰ (Fig. 4), which is higher than the amplitude (between 2.9 and 6.4 ‰) found in recently collected bivalves from Oubangui River (Kelemen et al., 2017). At the time of shell precipitation, $\delta^{13}C_{shell}$ values are derived from dissolved inorganic carbon (DIC) and the bivalve's own

metabolic (respired) carbon (McConnaughey et al., 1997; McConnaughey and Gillikin 2008), however the proportion is highly variable among species (McConnaughey et al., 1997; Lorrain et al., 2004; Gillikin et al., 2006, 2007; Poulain et al., 2010, Graniero et al., 2021). While young specimens during fast growth mainly use carbon from the DIC pool, older bivalves incorporate a relatively higher amount of metabolically derived carbon which is more [13]C-depleted than the DIC pool (Lorrain et al., 2004). In all analysed specimens, an ontogenetic decrease of $\delta^{13}C$ values was observed, which is in agreement with other

studies suggesting an inclination towards metabolic carbon incorporation in older age (Lorrain et al., 2004; Gillikin et al., 2006, 2007, 2009; McConnaughey and Gillikin, 2008). In the Oubangui basin, the weathering regime (silicate versus carbonate weathering) exerts strong control on $\delta^{13}CDIC$ values, but there is also a strong seasonality with higher values during low flow conditions, likely due to a combination increased aquatic primary production and $CO_2$ outgassing during low flow (Bouillon



et al., 2012). A reduction in precipitation could have an indirect effect on $\delta^{13}C_{shell}$ values by increasing $\delta^{13}C_{DIC}$ values of the
water during low flow, but this is not evident in the shell data. $\delta^{13}C_{shell}$ values did not closely match with $\delta^{13}C_{DIC}$ in the previous
study on *C. wissmanni* shells, thus they are not quantitative environmental proxies (Kelemen et al., 2017), however, they might
qualitatively indicate large scale changes in watershed characteristics (e.g., Goewert et al., 2007).

## 5. Conclusions

The analysis of $\delta^{18}O$ values in historical and contemporary freshwater bivalve shells from the Oubangui River demonstrated a
clear baseline shift in river $\delta^{18}O$ values over time. Shells collected between the late 19th and mid-20th century exhibited a
much narrower range of $\delta^{18}O$ values than recently collected shells. The seasonal cyclicity was nevertheless pronounced, and
our data indicate a higher intra-annual variation in river water $\delta^{18}O$ values in recent times, indicating a major shift in
hydroclimate baseline. The long-term discharge record shows a decline in both high and low discharge conditions, and when
discharge is reconstructed based on $\delta^{18}O$ data from historical shells, and a recently established $\delta^{18}O_w$-Q relationship, a
divergence in low flow periods suggests that a change in the $\delta^{18}O_w$-Q relationship might have occurred. The mechanisms
underlying such a shift are, however, not straightforward to identify. According to Nguimalet and Orange (2013, 2019), there
were only minor changes in precipitation patterns since 1935, and the $\delta^{18}O_{shell}$ values during high discharge-wet season
appeared to be similar in historical and recent shells, which suggests no substantial shift in (wet season) precipitation $\delta^{18}O$
values over the past century. However, the monthly precipitation data from Bangui and Bambari revealed a decrease in dry
season rainfall amounts, which should directly or indirectly influence the river water $\delta^{18}O$. The aquifer recharging capacity
above Bangui might be more limited since the severe drought in the 1970s. As it is mainly expected to recharge during abundant
rainfall (with low $\delta^{18}O$ values), the increased $\delta^{18}O$ values during low flow conditions in recent times coupled with significantly
decreased low discharge, would suggest that aquifers upstream of the rainforest belt did not yet completely recover their
capacity of supporting the river base flow.
Further studies of contemporary and past $\delta^{18}O_w$ patterns in precipitation across the basin (e.g., from direct precipitation
sampling and tree core $\delta^{18}O$ studies) should be able to provide more insight into changes in the hydroclimate of the Oubangui
basin and other such river basins for which no historical discharge data are available.

## Acknowledgements

We thank Didier van der Spiegel at the Royal Museum for Central Africa (MRAC), Yves Samyn at the Royal Belgian Institute
of Natural Sciences (KBIN), Philippe Bouchet and Virginie Héros at the MNHN Paris for providing access to museum
collections. We thank Dan Graf for help identifying the shell specimens and for his work on the MusselP database
(http://mussel-project.uwsp.edu), Thibault Lambert for producing Figure 1, and Anouk Verheyden for lab assistance. Funding
for this study was provided by the European Research Council (ERC Starting Grant StG 240002, AFRIVAL) to S.B., a National



Geographic Society Research and Exploration Grant (#8885-11) to D.P.G. and S.B., the KU Leuven Special Research Fund

(DBOF PhD scholarship to Z.K.), the Research Foundation Flanders (FWO-Vlaanderen; research project G.0D87.14N, and travel grants) to S.B. and D.P.G., and a Research Corporation for Science Advancement, Single-Investigator Cottrell College Science Award (#20169) to D.P.G.. The US National Science Foundation funded Union College's isotope ratio mass spectrometer and peripherals (NSF-MRI #1229258). We also thank Athanase Yambélé (Direction de la Météorologie Nationale, Bangui) who was instrumental in the collection of contemporary shells and data that made this follow-up study

possible, and to Gil Mahé (IRD, Montpellier, France) and Julien Thébault (Université de Bretagne Occidentale, Brest, France) for their constructive discussions on the data.

**Data availability:**

The full set of stable isotope data on historical shells is provided as an electronic supplement.

**Author contributions:**

This study was conceived by D.P.G. and S.B. All authors contributed to sample collection (recent and/or historical), sample analyses, and data interpretation. Z.K. drafted the initial version of the manuscript, which was revised by D.P.G. and S.B.

**Competing interests:**

One of the co-authors (S.B.) is an editorial board member of BG.

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



**Tables**

**Table 1. Data from all analyzed *C. wissmanni* valves including sample code, estimated downstream distance from Bangui, collection date and collection holder, and shell size.**

| Shell code | Museum | Collection (dd.mm.yyyy) | Estimated distance from Bangui (km) | Length (mm) | Width (mm) |
|---|---|---|---|---|---|
| 1891A | MNHN | 23.08.1891 | ~400 | 104 | 78 |
| 1891D | MNHN | 27.12.1891 | 0 | 100 | 72 |
| 1904 | IRSNB | 1904 | 100-250 | 55 | 35 |
| 1908 | MNHN | 15.04.1908 | ~400 | 87 | 59 |
| 1914A | IRSNB | 01.1914 | 0 | 79 | 52 |
| 1914B | IRSNB | 01.1914 | 0 | 96 | 59 |
| 1948 | MRAC | 1947-1948 | ~150 | 106 | 87 |
| 1950s | MNHN | ~1950s | ~230 | 109 | 71 |





**Figures**

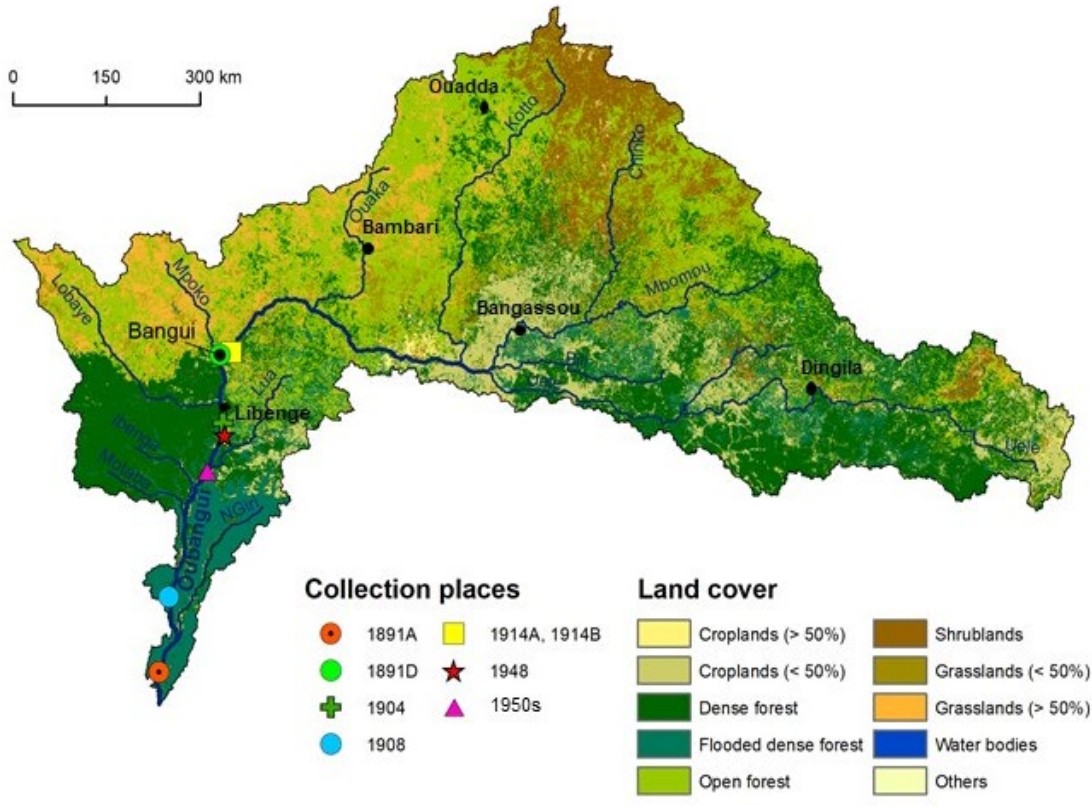


**Figure 1. The land cover of the Oubangui Basin and the most important tributaries. Different colored symbols represent the museum archived specimens at their collection place.**



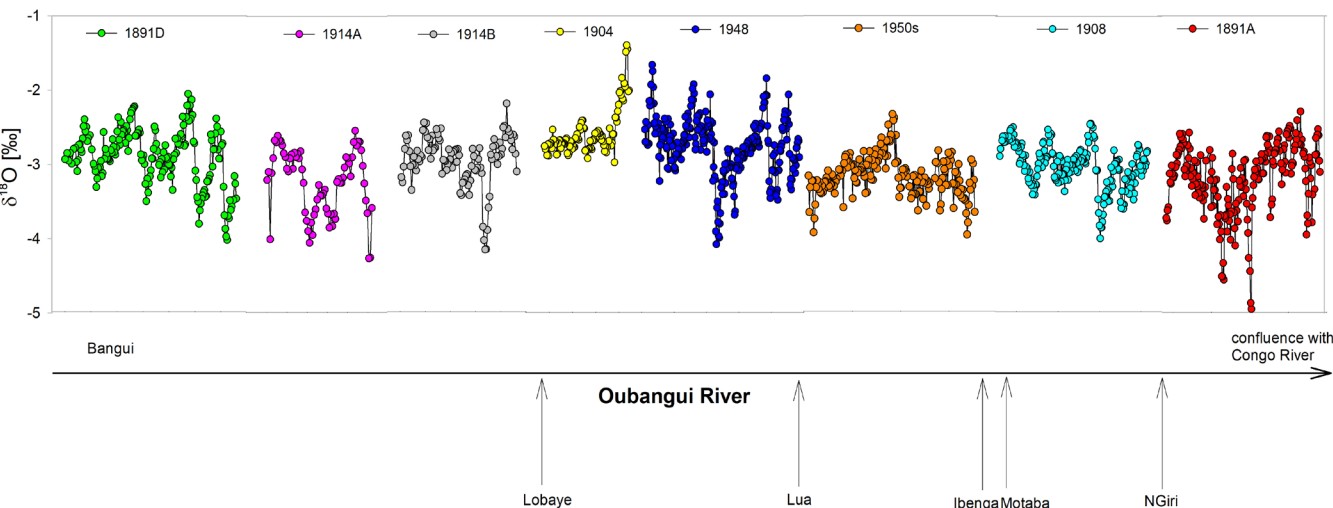

**Figure 2. δ¹⁸O values of all analyzed historical specimens in order of collection site, from Bangui towards the confluence with the Congo River. Arrows on the X-axis show the locations where important tributaries join to the Oubangui River. The year of shell collection is mentioned for each specimen on the symbol legend. See Table 1 for actual distance downstream from Bangui.**

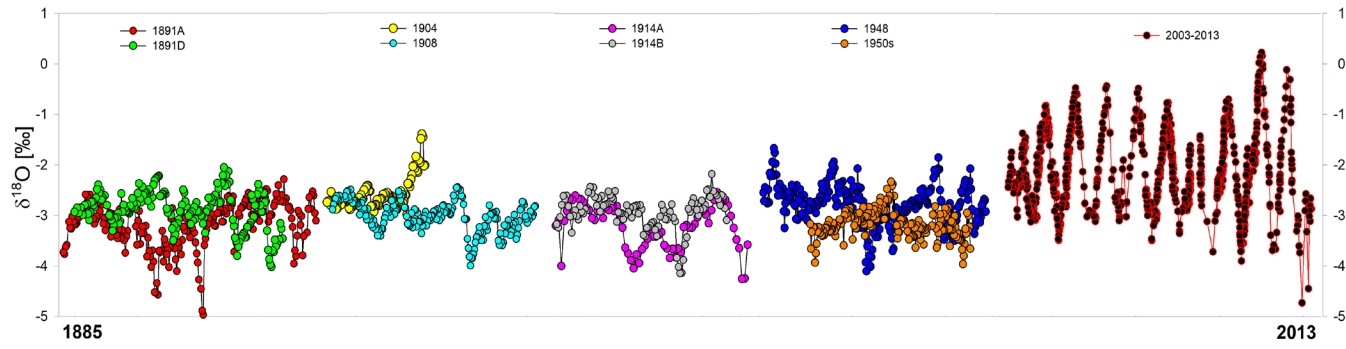

**Figure 3. δ¹⁸O_shell values of all analyzed archived and recently collected specimens in function of time, starting from the oldest archived specimens (collected in 1891) towards the most recently collected shells, where nine specimens are sorted chronologically into one master shell (recent shell data from Kelemen et al., 2017).**



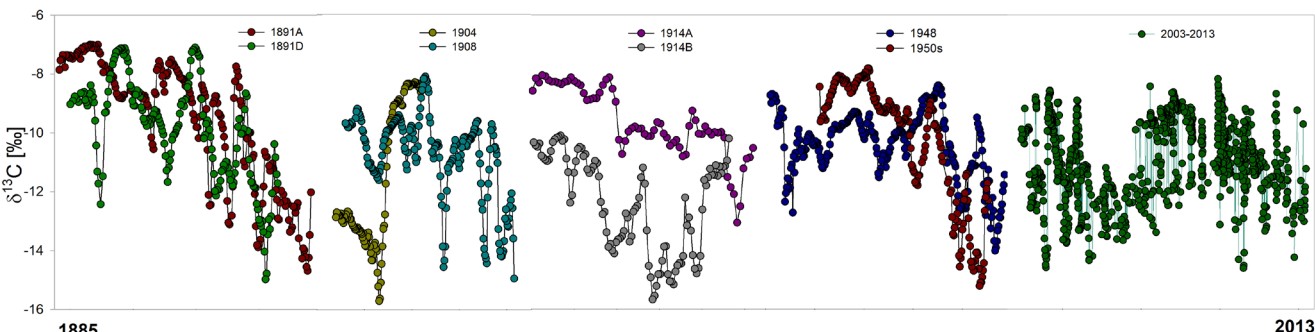

**Figure 4.** $\delta^{13}C_{shell}$ **values of bivalve shells vs time, starting from the oldest archived specimens (collected in 1891) towards the most recently collected shells, sorted chronologically into one master shell (recent shell data from Kelemen et al., 2017).**

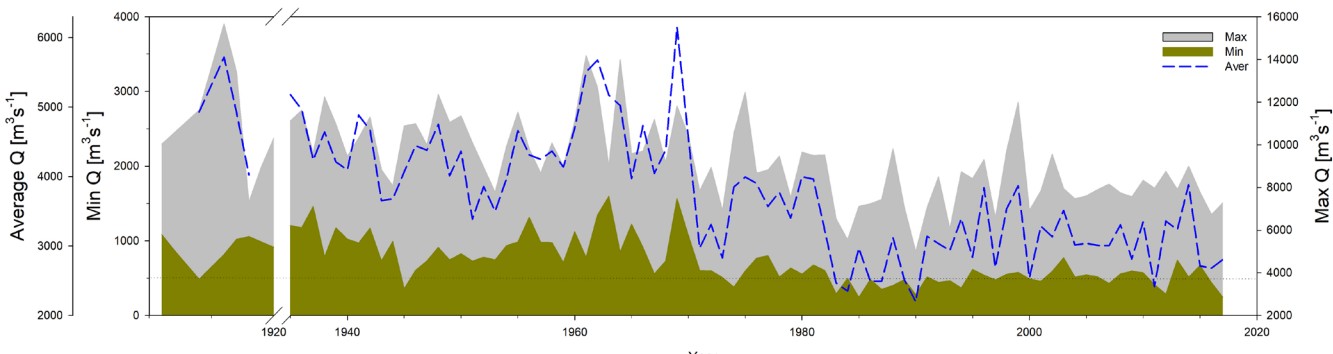


**Figure 5. Yearly maxima, minima and average values of the centennial discharge record. Yearly maxima and minima were calculated as the average of the 10 consecutive highest and lowest measured values each calendar year.**

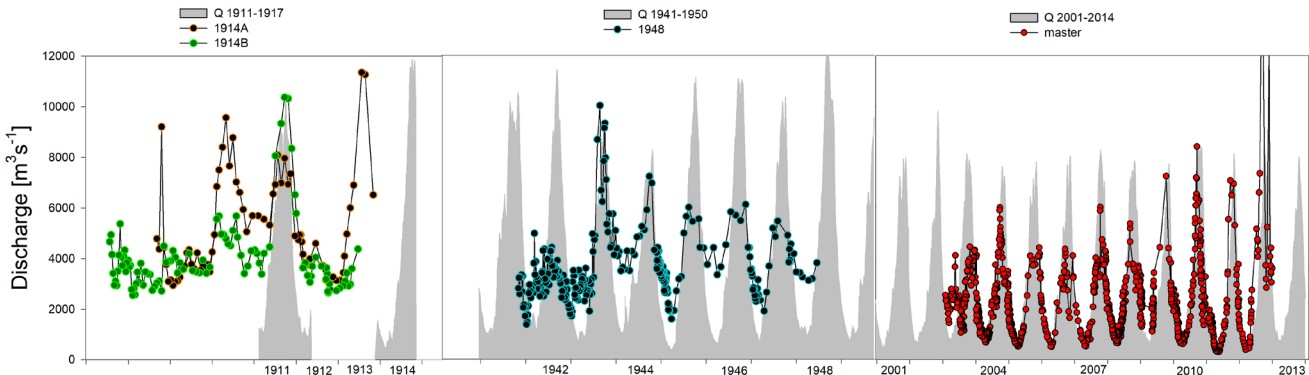

**Figure 6. Reconstructed discharge of the Oubangui River based on the shell $\delta^{18}O$ data and the logarithmic $\delta^{18}O_w$ - Q relationship measured for the period 2009-2013. The graphs focus on the shell data that coincide with the available discharge measurement and the grey area represents the measured discharge, while the colored symbols are discharge values reconstructed from $\delta^{18}O_{shell}$ values aligned with the best matching area.**





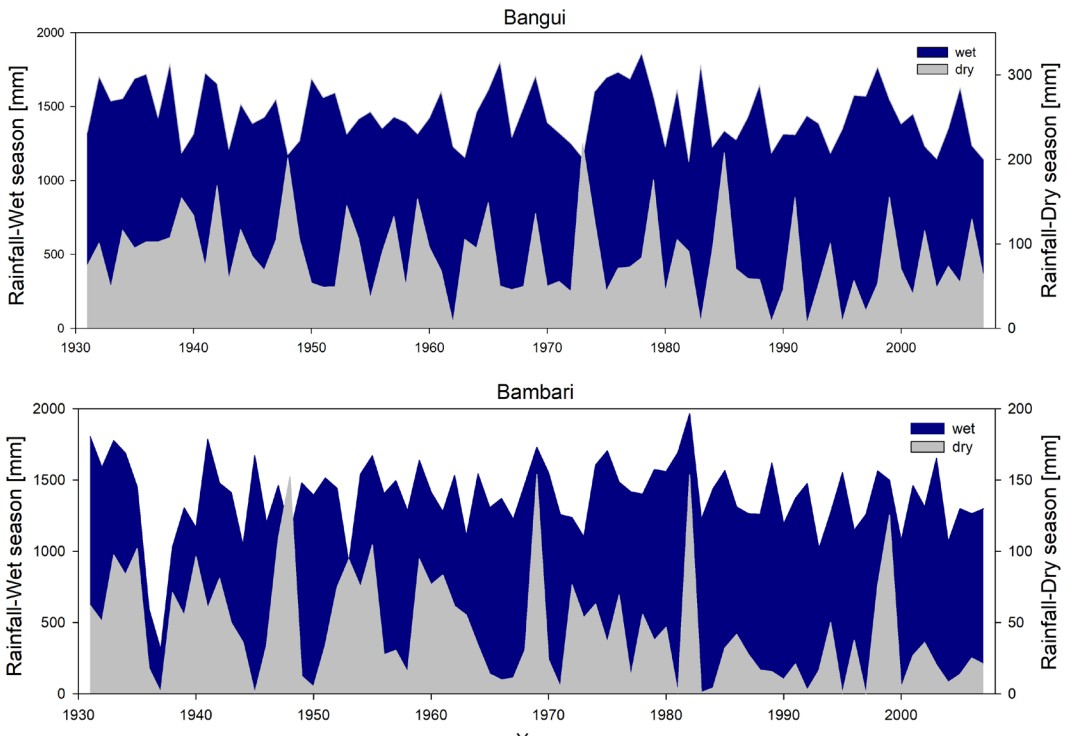

**630**  **Figure 7. Area plot of the dry (light area) and wet (dark area) season rainfall amount at Bangui (upper graph) and Bambari (lower graph) station. December, January and February are included in the dry season, while March to November represent the wet season.**

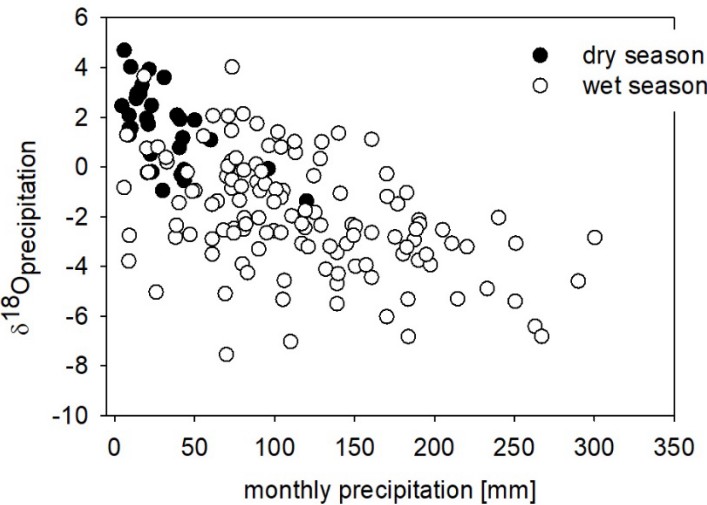

**Figure 8. Precipitation amount and δ$^{18}$O values at Bangui for the period 2009 – 2018 (two collection sites, data from the IAEA-GNIP database, 2 outliers were removed) Filled circles represent precipitation during the dry season (December, January, February) and**
**635**  **open circles the wet season (from March to November). The weighted average δ$^{18}$O value is -2.3 ‰.**