# Peer review of "Reconstructing Central African hydro-climate over the past century using freshwater bivalve shell geochemistry"

_EGUsphere, 2024_

## Author Comment (AC1)

We would like to thank both reviewers for the time taken to provide constructive and insightful comments on our manuscript. Below we have copied their original suggestions and comments in full, followed by a response to each of the issues raised and how these will be addressed in a revised version.

**RC2**: 'Comment on egusphere-2024-2714', Anonymous Referee #2, 17 Nov 2024

*REF:  This manuscript is well written, interesting, and has an easy to follow story. I think it absolutely deserves publication; well done authors! I have two suggestions that I think can improve the manuscript: I would suggest adding one or two sentences to the abstract or introduction explaining that higher oxygen isotope values are indicative of lower flow conditions. This relationship is key to understanding the premise and results of the paper, so I would recommend stating it explicitly, earlier in the paper. My second suggestion is to add more rationale in the method section as to why carbon isotopes were measured. Oxygen isotopes were clearly the focus, and it makes sense to also run carbon at the same time since the shell samples are already at the mass spec. But, I think more rational in the methods is needed for why carbon was included in the manuscript itself. I look forward to seeing this paper published.*

> Reply: Thanks for the very positive feedback on our work. We agree with the suggestion to mention the notion that higher $\delta^{18}O$ values correspond to low flow conditions early on (i.e. in the abstract). Secondly, we will also incorporate the suggested rationale for including $\delta^{13}C$ measurements. Indeed, on the one hand they are measured along with $\delta^{18}O$ data (hence, there are available by default), but there was also an inherent potential value in the $\delta^{13}C$ data, since we know from regular sampling on the Oubangui River that $\delta^{13}C$ values of dissolved inorganic carbon (DIC) follow a clear seasonality (Bouillon et al. 2012, 2014), and we would expect $\delta^{13}C$ values in bivalves shells to record this seasonality (although besides $\delta^{13}C$-DIC, metabolic $CO_2$ and other factors may also influence $\delta^{13}C_{carbonate}$ values). This will be made clear from the onset in the revised version.

---

## Author Comment (AC2)

We would like to thank both reviewers for the time taken to provide constructive and insightful comments on our manuscript. Below we have copied their original suggestions and comments in full, followed by a response to each of the issues raised and how these will be addressed in a revised version.

**RC1**: ['Comment on egusphere-2024-2714'](), Anonymous Referee #1, 12 Nov 2024

*REF:  General Comments*

*The authors present high resolution $\delta^{18}O$ and $\delta^{13}C$ measurements of bivalve shells collected from the Oubangui River basin in the Central African Republic to reconstruct hydroclimate since the late 1800s. Isotopic data from archival and recent shells are contextualized by instrumental measurements of river discharge and precipitation. This work appears to be a follow-up on an earlier study by these authors and others (Kelemen et al., 2017) where they separately assessed the equilibrium $\delta^{18}O_{carb}$-temperature-$\delta^{18}O_{water}$ relationship in this species and quantified a relationship between $\delta^{18}O_{water}$ and river discharge at this specific sample site. They observe an imperfect fit between the discharge estimated from shell $\delta^{18}O_{carb}$ using the aforementioned empirical relationship and the instrumentally measured discharge. They look to other observational records and conclude that the increased variability in $\delta^{18}O$ since 2003 is probably related to changes in dry season precipitation and groundwater flow.*

*Overall the dataset presented here is impressive, the work is technically sound in terms of methodology and interpretation, and the quality of writing is good. It furthermore addresses pertinent questions about central African hydroclimate. My comments should be considered minor and mostly aim to clarify certain points or draw out select points of discussion. The abundance of data considered here leads to very nuanced discussion of site-specific circumstances, which is all sound, but the paper doesn't circle back to the stated goal of evaluating the performance of bivalve $\delta^{18}O_{carb}$ as a more widely applicable proxy for discharge. I would like to see more discussion of this angle and more explicit consideration of limitations in other contexts.*

> Reply: We thank the reviewer for their overall positive evaluation of our work, and appreciate the constructive and insightful suggestions made below. We acknowledge that we did perhaps not pay sufficient attention to come back to our initial goals in the Discussion, and will expand on this in the revised Discussion.

**Specific comments**

*REF: Line 137-144: A clearer way to introduce these equations would be to first state that Equation 1 is used for calculating the aragonite-water fractionation factor based on Dettman et al. (1999). Subsequently, $\delta^{18}O_{water}$ was calculated using Equation 2, the alpha*

*value from Equation 1, and the embedded VPDB to VSMOW conversion. As an additional note: why not use a more recent calibration such as Kim et al. (2007) or Grossman (2012)? What is the effect of using these different equations on reconstructed $\delta^{18}O_{water}$ from the bivalves?*

Reply: We are open to restructuring this section as suggested. The Dettman et al. (1999) equation is based on the original Grossman and Ku (1986) equation, but re-written in 1000*ln$\alpha$ format. Dettman et al. tested this equation specifically on freshwater mussels, and has since become the standard equation used in the field, as evidenced by its high number of citations. Kim et al. (2007), in contrast, based their work on abiotic carbonate precipitation experiments under controlled conditions. Thus, will there are indeed several aragonite-specific equations in existence, most statistically indistinguishable, the Dettman et al. (1999) equation has become the most widely used, and is most relevant for freshwater mussels. To give an example of differences resulting from the choice of calibration, for a temperature and $\delta^{18}O_{shell}$ value within the range of our observations (28 °C and $\delta^{18}O_{shell}$ = -2‰), the Dettman et al. (1999) equation would result in a calculated $\delta^{18}O_{water}$ of -0.5‰, whereas the Kim et al. (2007) results in a calculated $\delta^{18}O_{water}$ of 0.2‰ (this difference is fairly constant across the range of conditions in our study). Given that relative variations are driving our conclusions, using a different equation would not modify their essence. Moreover, we provide the full dataset – should readers wish to use other calibrations, or updated equations emerge in the future that are deemed to be more appropriate, the necessary data to re-evaluate are available.

REF: *Line 210-213: Can the authors provide quantification or statistical tests for constancy in $\delta^{18}O$ minima to then contrast with the reported baseline shift post-1970?*

Reply: Interesting point, however - we prefer not to approach this statistically, the change in $\delta^{18}O$ minima is minor and variable, so not easy to interpret. We instead focus on the $\delta^{18}O$ maxima, which shows large differences in the modern shells (more than 1‰).

REF: *Line 226-227: "low discharge is overestimated" is a confusing turn of phrase here, particularly given the discussion of different effects on high vs low flow. Do the authors intend that overall discharge is simply underestimated (which seems to be what is represented in Fig. 6) or is there a component of low flow being referenced?*

Reply: We will try to avoid confusion by rephrasing this, but what the sentence should convey is that the reconstructed Q values (based on the recent Q-$\delta^{18}O_{water}$ relationship) is higher than measured Q values, in particular for the data during the 1940's (middle part of Figure 6) – this is not the case for recent shells (right part of Figure 6), there we do see a good match in dry season Q

between measured and reconstructed values. We will change the phrase to *"discharge during low flow is overestimated"*.

*REF:   Line 226-231: It does not seem accurate to say that the shells "consistently" underestimated Q for historical specimens. I count six years where the reconstructed Q matches peak measured Q impressively well, and the quality of the match doesn't seem to strictly depend on age of the record or size of the peak. What do the authors make of the variable accuracy of reconstructed Q, both before and after the hydrologic shift in low-flow water sources post-1970? They seem to suggest growth cessation during high discharge could account for this based on previous results (Kelemen et al., 2017), But how would this account for excellent accuracy during some of the highest peaks in the historical record? This is an important element of the discussion as the authors are setting this up as an evaluation of the proxy for future applications where less/no historical data is available.*

> Reply: We are not fully sure where the reviewer observes six years where peak Q data match well with reconstructed values/estimates in the older shells (pre-2001). We see matches for 3 wet seasons (1911, 1943, and 1944), and for one dry season (early 1942), for other dry and wet seasons there is considerable offset. Nevertheless, point taken that we should not refer to this as 'consistently', and we will rephrase accordingly and add some discussion on this. We indeed offered growth cessation as a possible mechanism to explain why peak discharge may not be captured by the shell $\delta^{18}O$ data, but it should be noted that there is considerable scatter in the water $\delta^{18}O$ data for high Q (see Kelemen et al. 2021) – we will add some notes on this as an additional potential caveat.

*REF:  Line 255: The authors state that a 5% overall reduction in rainfall post-1970 was only 5% cannot account for the 22% decrease in Q; they should quantitatively evaluate the significance of the change in dry season precipitation if they believe this to be a stronger causal mechanism.*

> Reply: Nguimalet and Orange (2019) had concluded that the decrease in rainfall was not enough to account for the decrease in discharge. We will provide a more quantitative interpretation of the changes in the dry season precipitation in the revised version.

*REF:  Line 317-318: This sentence is confusing—do the authors mean to refer to historical specimens twice?*

> Reply: Indeed, this is an error, should have read "in comparison with the $\delta^{18}O_{shell}$ values in recent specimens".

*REF:  Conclusions: The paper is pitched as a broad evaluation of bivalve shell $\delta^{18}O$ as an archive of various hydroclimate parameters, especially seasonal discharge but also precipitation dynamics, land cover, groundwater flow, and geography. Given the*

*limitations and enmeshed signals that they explore throughout the discussion, the authors should end with an overarching conclusion about how this proxy will best be used and interpreted in other systems. Some guiding questions: Their assessment of possible changes in precipitation dynamics, land cover, groundwater flow etc. based on $\delta^{18}O_{water}$ follow on from identification of a $\delta^{18}O_{water}$-Q mismatch. Could these factors be evaluated without an established $\delta^{18}O_{water}$-Q relationship (which would apply to any other river system and to historical systems without direct monitoring)? Could they be evaluated in a context where Q is indeed a direct control on $\delta^{18}O_{water}$? How would one identify when an empirical $\delta^{18}O_{water}$-Q relationship does break down in the past, as argued here for the Oubangui, without direct Q data for comparison? How far into the past could a $\delta^{18}O$-Q relationship measured recently be extended into the past even for the same river basin?*

> Reply: We acknowledge that our Discussion would benefit from getting back to the objectives set out at the start of the manuscript; and appreciate the guiding questions proposed. While we do not have definite answers to all of the questions raised, we agree that it's a good idea to make full circle and to add some critical thoughts on these issues in the revised Discussion.

*REF:  Figure 1: An inset map showing the basin's footprint in the CAR with a lat/long grid would be appropriate to ground the reader in absolute space. Some of the text in the lower left is difficult to read against the dark background. Symbols could be slightly bigger.*

> Reply: We will try to improve the map, showing the outline of the basin within the Central African Republic, and the broader Congo Basin. We will also try to improve the visibility of the text.

*REF:  Figure 2: A mixture of color and symbol could be used to differentiate overlapping records. Are the overlapping records matched absolutely in time based on collection date/band counting, matched based on features of the profile, or somewhat arbitrarily overlaid?*

> Reply: The records are represented in sample space along the shell, without data point adjustment leading to somewhat of an arbitrary x-axis. Thus, the x-axis should not be considered an absolute time axis, as the goal of Figure 2 was to show the possible effect of different rivers on the relative $\delta^{18}O_{shell}$ values.

*REF:  Figure 3: Same comment as Fig. 2.*

> Reply: We will improve this Figure by using different symbols. Regarding the overlap, shells are "arbitrarily" overlaid in sample collection date batches; see response to previous suggestion.

*REF:  Figure 6: Same comment as Fig. 2.*

Reply:   see response to previous suggestion.

**Technical corrections**

*REF:  Throughout: be consistent with use of Q as an abbreviation for discharge.*

Reply: We will go through the text to ensure we consistently use this abbreviation.

*REF:  Throughout: The authors use the term "historical" as a contrast to "recent" shells throughout, but "historical" is used to group different subsets of shells in different parts of the discussion (e.g. pre-1960, pre-1970). It would be clearer to simply refer to groups of shells by their specific time periods since the collection dates are well-constrained.*

Reply: We will carefully go through our use of the term "historical" throughout the manuscript and see where we can be more specific, while reserving the term to refer to all non-recent shell material.

*REF:  Line 77: "Predicted" would be more apt than "hypothesized" here.*

Reply: This will be replaced in the revised version.

*REF:  Line 84: superscript $m^3s^{-1}$*

Reply: This will be corrected in the revised version.

*REF:  Line 135: subscript on $CaCO_3$*

Reply: This will be corrected in the revised version.

*REF:  Line 177: Cite Figure 3.*

Reply:  Reference will be made to Figure 3 here in the revised version.

*REF:  Line 189: Cite Figure 5.*

Reply:  Figure 5 is cited in this sentence  (on L190) ?

*REF:  Line 195: superscript on $\delta^{18}O$*

Reply: This will be corrected in the revised version.

*REF:  Line 233: spacing on $\delta^{18}O_{water}$-Q.*

Reply: This will be corrected in the revised version.

*REF:  Line 249: only in 1970 or beginning in 1970?*

Reply: from 1970 onwards, this was indeed not clear – we will make this explicit.

*REF:  Line 159: superscript on $\delta^{18}O$*

Reply: This will be corrected in the revised version.

*REF:  Line 267: better to say "All these factors could lead..."*

Reply: This will be corrected in the revised version.

*REF:  Line 279: superscript on $m^3s^{-1}$*

Reply: This will be corrected in the revised version.

*REF:  Line 279: "Qmax/Qmin of about..."*

Reply: This will be corrected in the revised version.

*REF:  Line 280: superscript on $\delta^{18}O_{water}$*

Reply: This will be corrected in the revised version.

*REF:  Line 340: subscript on $\delta^{13}C_{DIC}$*

Reply: This will be corrected in the revised version.

---

## Author Response (AR1)

**Author replies to editorial and reviewer comments**

Below, we have copied the comments and suggestions of the two reviewers, and each point raised is followed by a statement on how it was addressed in the revised version of the manuscript.

**RC1**: 'Comment on egusphere-2024-2714', Anonymous Referee #1, 12 Nov 2024

*REF: General Comments*

*The authors present high resolution $\delta^{18}O$ and $\delta^{13}C$ measurements of bivalve shells collected from the Oubangui River basin in the Central African Republic to reconstruct hydroclimate since the late 1800s. Isotopic data from archival and recent shells are contextualized by instrumental measurements of river discharge and precipitation. This work appears to be a follow-up on an earlier study by these authors and others (Kelemen et al., 2017) where they separately assessed the equilibrium $\delta^{18}O_{carb}$-temperature-$\delta^{18}O_{water}$ relationship in this species and quantified a relationship between $\delta^{18}O_{water}$ and river discharge at this specific sample site. They observe an imperfect fit between the discharge estimated from shell $\delta^{18}O_{carb}$ using the aforementioned empirical relationship and the instrumentally measured discharge. They look to other observational records and conclude that the increased variability in $\delta^{18}O$ since 2003 is probably related to changes in dry season precipitation and groundwater flow.*

*Overall the dataset presented here is impressive, the work is technically sound in terms of methodology and interpretation, and the quality of writing is good. It furthermore addresses pertinent questions about central African hydroclimate. My comments should be considered minor and mostly aim to clarify certain points or draw out select points of discussion. The abundance of data considered here leads to very nuanced discussion of site-specific circumstances, which is all sound, but the paper doesn't circle back to the stated goal of evaluating the performance of bivalve $\delta^{18}O_{carb}$ as a more widely applicable proxy for discharge. I would like to see more discussion of this angle and more explicit consideration of limitations in other contexts.*

> Reply: We thank the reviewer for their overall positive evaluation of our work, and appreciate the constructive and insightful suggestions made below. We acknowledge that we did perhaps not pay sufficient attention to come back to our initial goals in the Discussion, and expanded on this in the revised Discussion.

**Specific comments**

*REF: Line 137-144: A clearer way to introduce these equations would be to first state that Equation 1 is used for calculating the aragonite-water fractionation factor based on*

*Dettman et al. (1999). Subsequently, $\delta^{18}O_{water}$ was calculated using Equation 2, the alpha value from Equation 1, and the embedded VPDB to VSMOW conversion. As an additional note: why not use a more recent calibration such as Kim et al. (2007) or Grossman (2012)? What is the effect of using these different equations on reconstructed $\delta^{18}O_{water}$ from the bivalves?*

> Reply: The Dettman et al. (1999) equation is based on the original Grossman and Ku (1986) equation, but re-written in 1000*lnα format. Dettman et al. tested this equation specifically on freshwater mussels, and has since become the standard equation used in the field, as evidenced by its high number of citations. Kim et al. (2007), in contrast, based their work on abiotic carbonate precipitation experiments under controlled conditions. Thus, will there are indeed several aragonite-specific equations in existence, most statistically indistinguishable, the Dettman et al. (1999) equation has become the most widely used, and is most relevant for freshwater mussels.  To give an example of differences resulting from the choice of calibration, for a temperature and $\delta^{18}O_{shell}$ value within the range of our observations (28 °C and  $\delta^{18}O_{shell}$ = -2‰), the Dettman et al. (1999) equation would result in a calculated $\delta^{18}O_{water}$ of -0.5‰, whereas the Kim et al. (2007) results in a calculated $\delta^{18}O_{water}$ of 0.2‰ (this difference is fairly constant across the range of conditions in our study). Given that relative variations are driving our conclusions, using a different equation would not modify their essence.

> Corrections have been made according to the suggestions (see Lines 157-164). We decided, however, to retain the Dettman et al. (1999) equation, as it is the most closely adapted to freshwater mussels and most widely used equation (not to mention it being statically indistinguishable from the others).

*REF:  Line 210-213: Can the authors provide quantification or statistical tests for constancy in $\delta^{18}O$ minima to then contrast with the reported baseline shift post-1970?*

> Reply: Interesting point, however - we prefer not to approach this statistically, the change in $\delta^{18}O$ minima is minor and variable, so not easy to interpret. We instead focus on the $\delta^{18}O$ maxima, which shows large differences in the modern shells (more than 1‰).

*REF:  Line 226-227: "low discharge is overestimated" is a confusing turn of phrase here, particularly given the discussion of different effects on high vs low flow. Do the authors intend that overall discharge is simply underestimated (which seems to be what is represented in Fig. 6) or is there a component of low flow being referenced?*

> Reply: The phrase *"low discharge is overestimated"* has been changed to *"discharge during low flow is overestimated".*

*REF:   Line 226-231: It does not seem accurate to say that the shells "consistently" underestimated Q for historical specimens. I count six years where the reconstructed Q matches peak measured Q impressively well, and the quality of the match doesn't seem to strictly depend on age of the record or size of the peak. What do the authors make of the variable accuracy of reconstructed Q, both before and after the hydrologic shift in low-flow water sources post-1970? They seem to suggest growth cessation during high discharge could account for this based on previous results (Kelemen et al., 2017), But how would this account for excellent accuracy during some of the highest peaks in the historical record? This is an important element of the discussion as the authors are setting this up as an evaluation of the proxy for future applications where less/no historical data is available.*

> Reply: We are not fully sure where the reviewer observes six years where peak Q data match well with reconstructed values/estimates in the older shells (pre-2001). We see matches for 3 wet seasons (1911, 1943, and 1944), and for one dry season (early 1942), for other dry and wet seasons there is considerable offset. Nevertheless, point taken that we should not refer to this as 'consistently'.  We changed this to: "which indeed indicates that discharge during low flow is overestimated in the older shells (Fig. 6)." Line 266. Fig 6 shows that all older shells overestimated low flow conditions.

*REF:  Line 255: The authors state that a 5% overall reduction in rainfall post-1970 was only 5% cannot account for the 22% decrease in Q; they should quantitatively evaluate the significance of the change in dry season precipitation if they believe this to be a stronger causal mechanism.*

> Reply: Nguimalet and Orange (2019) had concluded that the decrease in rainfall was not enough to account for the decrease in discharge. A sentence with a more quantitative interpretation has been added, providing the values for the precipitation decrease at Bambari and Bangui, the two stations discussed in the manuscript.

*REF:  Line 317-318: This sentence is confusing—do the authors mean to refer to historical specimens twice?*

> Reply: Indeed, this is an error, should have read "in comparison with the $\delta^{18}O_{shell}$ values in recent specimens".    The sentence has been changed accordingly.

*REF:  Conclusions: The paper is pitched as a broad evaluation of bivalve shell $\delta^{18}O$ as an archive of various hydroclimate parameters, especially seasonal discharge but also precipitation dynamics, land cover, groundwater flow, and geography. Given the limitations and enmeshed signals that they explore throughout the discussion, the authors should end with an overarching conclusion about how this proxy will best be used and interpreted in other systems. Some guiding questions: Their assessment of possible changes in precipitation dynamics, land cover, groundwater flow etc. based on*

*$\delta^{18}O_{water}$ follow on from identification of a $\delta^{18}O_{water}$-Q mismatch. Could these factors be evaluated without an established $\delta^{18}O_{water}$-Q relationship (which would apply to any other river system and to historical systems without direct monitoring)? Could they be evaluated in a context where Q is indeed a direct control on $\delta^{18}O_{water}$? How would one identify when an empirical $\delta^{18}O_{water}$-Q relationship does break down in the past, as argued here for the Oubangui, without direct Q data for comparison? How far into the past could a $\delta^{18}O$-Q relationship measured recently be extended into the past even for the same river basin?*

Reply: While we do not have definite answers to all of the questions raised, we agree that it's a good idea to make full circle and to add some critical thoughts on these issues in the revised Discussion. We added the caveat at the end of the conclusion: "While we have previously shown that tropical bivalves living in rivers with near-constant annual water temperatures faithfully record water $\delta^{18}O$ values in their shells, this study shows that interpreting the $\delta^{18}O_w$ values using the $\delta^{18}O_w$-Q relationship is more complex as several parameters can influence this relationship. Nevertheless, large changes in river discharge over the past century were clearly reflected in shell $\delta^{18}O$ values. We posit that archived collections of freshwater shells can provide a wealth of hydroclimate proxy data over the past century.

REF: *Figure 1: An inset map showing the basin's footprint in the CAR with a lat/long grid would be appropriate to ground the reader in absolute space. Some of the text in the lower left is difficult to read against the dark background. Symbols could be slightly bigger.*

Reply: Following these suggestions, the map has been improved. We added a latitude/longitude grid to the main map and included a small map of the region in the upper right corner. The visibility was enhanced by adjusting certain colors to better contrast with the background, and the symbols were modified and enlarged.

REF: *Figure 2: A mixture of color and symbol could be used to differentiate overlapping records. Are the overlapping records matched absolutely in time based on collection date/band counting, matched based on features of the profile, or somewhat arbitrarily overlaid?*

Reply: The records are represented in sample space along the shell, without data point adjustment leading to somewhat of an arbitrary x-axis. Thus, the x-axis should not be considered an absolute time axis, as the goal of Figure 2 was to show the possible effect of different rivers on the relative $\delta^{18}O_{shell}$ values. Figure 2 has been modified: the symbols have been changed to triangles, circles, and squares. A unique symbol is assigned to each individual shell, which is used consistently in all subsequent figures. We have updated the figure caption to make the positioning of the shells clearer.

*REF:  Figure 3: Same comment as Fig. 2.*

Reply: We improved this Figure by using different symbols. Regarding the overlap, shells are "arbitrarily" overlaid in sample collection date batches; see response to previous suggestion.

*REF:  Figure 6: Same comment as Fig. 2.*

Reply: See response to Fig. 2. Regarding the overlap, only two shells overlap on this figure (1914 shells), and they are distinct from one another.

**Technical corrections**

*REF:  Throughout: be consistent with use of Q as an abbreviation for discharge.*

Reply: We have gone through the text to ensure we consistently use this abbreviation.

*REF:  Throughout: The authors use the term "historical" as a contrast to "recent" shells throughout, but "historical" is used to group different subsets of shells in different parts of the discussion (e.g. pre-1960, pre-1970). It would be clearer to simply refer to groups of shells by their specific time periods since the collection dates are well-constrained.*

Reply: We have carefully gone through our use of the term "historical" throughout the manuscript; but found no instances where the term was used to refer to a specific time period of the older shells. On L. 80, we mention the term refers to pre-1960 shells.

*REF:  Line 77: "Predicted" would be more apt than "hypothesized" here.*

Reply: This has been replaced.

*REF:  Line 84: superscript $m^3s^{-1}$*

Reply: This has been corrected.

*REF:  Line 135: subscript on $CaCO_3$*

Reply: This has been corrected.

*REF:  Line 177: Cite Figure 3.*

Reply:  This has been corrected.

*REF:  Line 189: Cite Figure 5.*

Reply:  Figure 5 is already cited at the end of this sentence.

*REF:  Line 195: superscript on $\delta^{18}O$*

Reply: This has been corrected.

*REF:  Line 233: spacing on $\delta^{18}O_{water}$-Q.*

Reply: This has been corrected.

*REF:  Line 249: only in 1970 or beginning in 1970?*

Reply: The sentence has been corrected to more explicitly refer to the period starting from 1970.

*REF:  Line 159: superscript on $\delta^{18}O$*

Reply: This has been corrected.

*REF:  Line 267: better to say "All these factors could lead..."*

Reply: This has been corrected.

*REF:  Line 279: superscript on $m^3s^{-1}$*

Reply: This has been corrected.

*REF:  Line 279: "Qmax/Qmin of about..."*

Reply: This has been corrected.

*REF:  Line 280: superscript on $\delta^{18}O_{water}$*

Reply: This has been corrected.

*REF:  Line 340: subscript on $\delta^{13}C_{DIC}$*

Reply: This has been corrected.

**RC2**: 'Comment on egusphere-2024-2714', Anonymous Referee #2, 17 Nov 2024

*REF:  This manuscript is well written, interesting, and has an easy to follow story. I think it absolutely deserves publication; well done authors! I have two suggestions that I think can improve the manuscript: I would suggest adding one or two sentences to the abstract or introduction explaining that higher oxygen isotope values are indicative of lower flow conditions. This relationship is key to understanding the premise and results of the paper, so I would recommend stating it explicitly, earlier in the paper. My second suggestion is to add more rationale in the method section as to why carbon isotopes were measured. Oxygen isotopes were clearly the focus, and it makes sense to also run carbon at the same time since the shell samples are already at the mass spec. But, I think more rational in the methods is needed for why carbon was included in the manuscript itself. I look forward to seeing this paper published.*

Reply: Thanks for the very positive feedback on our work. A sentence has been added to the abstract explaining the relationship between discharge and oxygen isotope values.  We also added a sentence to the introduction (L79) explaining why we are also analyzing C isotopes.